# Is Intrinsic Cardioprotection a Laboratory Phenomenon or a Clinically Relevant Tool to Salvage the Failing Heart?

**DOI:** 10.3390/ijms242216497

**Published:** 2023-11-18

**Authors:** Tanya Ravingerova, Adriana Adameova, Lubomir Lonek, Veronika Farkasova, Miroslav Ferko, Natalia Andelova, Branislav Kura, Jan Slezak, Eleftheria Galatou, Antigone Lazou, Vladislava Zohdi, Naranjan S. Dhalla

**Affiliations:** 1Institute for Heart Research, Centre of Experimental Medicine, Slovak Academy of Sciences, 9 Dubravska cesta, 841 04 Bratislava, Slovakia; adriana.duris.adameova@uniba.sk (A.A.); lubomir.lonek@savba.sk (L.L.); veronika.farkasova@savba.sk (V.F.); miroslav.ferko@savba.sk (M.F.); natalia.andelova@savba.sk (N.A.); branislav.kura@savba.sk (B.K.); jan.slezak@savba.sk (J.S.); 2Department of Pharmacology and Toxicology, Faculty of Pharmacy, Comenius University in Bratislava, 10 Odbojárov St., 832 32 Bratislava, Slovakia; 3School of Biology, Aristotle University of Thessaloniki, 541 24 Thessaloniki, Greece; eleftheriagala@gmail.com (E.G.); lazou@bio.auth.gr (A.L.); 4Department of Life and Health Sciences, University of Nicosia, 2417 Nicosia, Cyprus; 5Department of Anatomy, Faculty of Medicine, Comenius University in Bratislava, 24 Špitalska, 813 72 Bratislava, Slovakia; vladislava.zohdi@monash.edu; 6Department of Anatomy and Developmental Biology, Monash Biomedicine Discovery Institute, Monash University, 19 Innovation Walk, Clayton, VIC 3800, Australia; 7Institute of Cardiovascular Sciences St. Boniface Hospital Albrechtsen Research Centre, 351 Tache Avenue, Winnipeg, MB R2H 2A6, Canada; nsdhalla@sbrc.ca

**Keywords:** ischemia/reperfusion injury, heart failure, adaptation, remote preconditioning, exercise-induced preconditioning, protective cell signaling

## Abstract

Cardiovascular diseases, especially ischemic heart disease, as a leading cause of heart failure (HF) and mortality, will not reduce over the coming decades despite the progress in pharmacotherapy, interventional cardiology, and surgery. Although patients surviving acute myocardial infarction live longer, alteration of heart function will later lead to HF. Its rising incidence represents a danger, especially among the elderly, with data showing more unfavorable results among females than among males. Experiments revealed an infarct-sparing effect of ischemic “preconditioning” (IPC) as the most robust form of innate cardioprotection based on the heart’s adaptation to moderate stress, increasing its resistance to severe insults. However, translation to clinical practice is limited by technical requirements and limited time. Novel forms of adaptive interventions, such as “remote” IPC, have already been applied in patients, albeit with different effectiveness. Cardiac ischemic tolerance can also be increased by other noninvasive approaches, such as adaptation to hypoxia- or exercise-induced preconditioning. Although their molecular mechanisms are not yet fully understood, some noninvasive modalities appear to be promising novel strategies for fighting HF through targeting its numerous mechanisms. In this review, we will discuss the molecular mechanisms of heart injury and repair, as well as interventions that have potential to be used in the treatment of patients.

## 1. Introduction

Cardiovascular diseases (CVDs) are the leading cause of death worldwide; it is estimated that, by 2030, approximately 24 million people will die from CVDs [1]. Although the advances in the prevention, diagnosis, and management of CVDs have progressed in the past three decades, chronic ischemic heart disease (IHD) and acute myocardial infarction (AMI) are often followed by the development of heart failure (HF) and are the major causes of death and morbidity in the developed world [2]. In Europe, more than 14 million people suffer from HF, and the number of cases is increasing (300,000 per year) [1]. The expected growth in the elderly population is proposed to be associated with a significant increase (50% in the USA over the next 15 years) in the number of patients diagnosed with HF [3]. Since the average life expectancy is constantly increasing not only in Western society but also in the developing world, HF will remain a significant health problem, with the five-year mortality still worse than that of many cancers [4]. Potential reasons for this may be related to the complexity of the mechanisms of the HF and the requirement of the multi-targeting interventions for optimal cardioprotection [5,6]. Early restoration of blood flow in the ischemic myocardium is an inevitable prerequisite of successful heart salvage. However, revascularization may paradoxically induce ischemia/reperfusion (I/R) injury and accelerate cardiomyocytes death, leading to increased size of infarction and reduced heart function, further progressing into HF [7]. Currently, data on forceful and reliable prevention of AMI and on ultimate reduction in I/R injury are unavailable. Therefore, there is still an unmet need to search for alternative strategies based on novel approaches that could specifically address repair and regeneration of the damaged and/or lost myocardium (given limited endogenous repair of cardiomyocytes in adults). In this respect, a phenomenon of ischemic pre- and postconditioning, as the most robust forms of innate cardioprotection observed in all animal species, including humans [8,9], as well as other forms of conditioning interventions appear to have potential in clinical conditions; this requires detailed study of their molecular mechanisms [5].

## 2. Development of Heart Failure

Numerous pathological processes in the heart end up with an injury of cardiomyocytes and decreased myocardial contractility in the long-term. IHD and AMI are the most severe diseases in this respect. Due to the progress in the management of CVD, including bypass surgery of AMI and an increasing number of heart transplantations, the patients live longer. However, the consequences of the initial heart injury, such as a long-lasting reduction in its contractile function, will later lead to the failure of the heart [10]. 

To a large extent, in different forms of cardiomyopathies, such as diabetic cardiomyopathy, oxidative stress and Ca^2+^ overload play a crucial role in subcellular remodeling and functional disorders [11,12]. Moreover, adverse effects of therapy of various types of cancers with anthracyclines (e.g., with doxorubicin) or radiation therapy induce cardiotoxicity related to oxidative stress [13,14,15]. In addition, valve diseases and hypertension-induced cardiac hypertrophy, among other factors, contribute to the development of chronic HF [3].

Importantly, although heart transplantation has become a routine treatment method, I/R damage to the heart after cold ischemic storage and subsequent reperfusion with warm oxygenated blood is critical in restoring heart function due to increased reactive oxygen species (ROS) production [16]. I/R injury in the donor heart has been found to be associated with an inflammatory response and apoptosis, aggravating myocardial function [17].

### 2.1. Severity of Heart Failure Is Sex- and Aging-Related 

The incidence of cardiac ailments differs with gender and age [18,19,20,21]. Aging itself leads to a decline in heart function, even among middle-aged persons without HF, due to a loss of cardiomyocytes and reactive cellular hypertrophy [22]. Among the elderly population, a dramatic loss of cells and enlargement of the remaining myocytes may represent a structural basis for cardiac remodeling, leading to heart dysfunction and failure [23]. Unfavorable effects of aging differ in males and females. Thus, it has been demonstrated in several epidemiological studies that females aged between 55 and 64 years are much less susceptible to the incidence of HF than males of the same age [20]. However, after the age of 65 years, the risk of HF increases more in females than in males [24]. It has been thus proposed that, among pre-menopausal women, ovarian hormones protect their hearts against CVD in general [25], and it is the gradual decline in these hormones that is responsible for the faster development of the lifestyle risk factors contributing to the higher risk of HF. There is a need for proper management of AMI in females to prevent late hospital attendance and delayed onset of therapy, as well as other adverse effects that may be involved in the development of HF [26]. Furthermore, many studies (both clinical and experimental) are currently focused on investigating cardioprotective interventions, mainly among healthy adult male subjects. However, sex-dependent differences are involved in the different responses of the heart to I/R injury in males and females and subsequent preservation of their heart function, especially in the elder population, and that should be considered [27]. Our recent studies also demonstrated gender-related differences with respect to ischemic tolerance that started to be obvious in juvenile rats [28,29,30].

### 2.2. Lifestyle Risk Factors

Risk factors associated with modern lifestyle, such as hypertension, chronic stress, diabetes, hyperglycemia, obesity, and dyslipidemia, have all been shown to have a negative impact on myocardial response to ischemia, to attenuate the effect of protective interventions and to accelerate thus the progression of HF [31,32,33,34,35,36,37,38]. It is interesting that chronic hypertension results in the hypertrophy of the left ventricle, and this was found to have a detrimental effect on cardiac function under basal pre-ischemic conditions [39].

## 3. Pathophysiological Mechanisms of Heart Failure

The pathophysiological mechanisms of HF are complex and heterogeneous, including processes such as hemodynamic overload, fibrosis, inflammation, endothelial dysfunction, ventricular remodeling, altered gene expression and Ca^2+^ cycling, oxidative stress, acceleration of apoptosis, necrosis, necroptosis, ferroptosis, mitochondrial dysfunction, and loss of cardiac cells in various cell death models [11,40,41,42].

### Cell Death Mechanisms

Many conventional (autophagy, necrosis, and apoptosis) and less-known cell death modalities, such as necroptosis [43,44] and pyroptosis [45,46], have been identified in various types of HF. Nowadays, it is widely accepted that non-apoptotic cell death modes, rather than apoptosis [47,48], underlie, at least in part, the altered cardiac function and geometry of the left ventricle. Likewise, autophagy, particularly a maladaptive form, seems to play an essential role in HF pathophysiology, although the relevance of this cell death mode might be significantly determined by its model/etiology [49,50,51,52]. Moreover, cell loss in HF is not limited just by one cell death mode, and its various types can co-exist independently of each other, and/or one type precedes another one, indicating that their signaling is activated concomitantly and/or sequentially. Furthermore, a link between autophagy and necroptosis has been demonstrated in both acute and chronic ischemia [50,52,53]. However, in a model of post-infarction HF, mimicking the New York Heart Association (NYHA) II-III stage of HF, autophagy was unlikely activated while necroptosis was present. In fact, canonical necroptosis signaling involving the RIP3-MLKL (receptor-interacting protein kinase 3- mixed lineage kinase domain-like protein) axis, has been documented in the infarcted area accompanied by increased levels of TNF (tumor necrosis factor) and activation of the caspase-1-IL-1β axis. This pro-inflammatory downstream pathway of RIP3, along with the lower caspase-8 levels, has also been detected in the non-infarcted zone of failing rat hearts [44]. Thus, RIP3 activation in the infarcted area leading to the disruption of the plasma membrane can induce the release of intracellular content and pro-inflammatory mediators to further amplify a diffused pro-inflammatory response, indicating a sequential order of these closely related cellular events. In addition, RIP3, as a convergence point of multiple signaling pathways, may play an important role in the process of I/R. On the other hand, it was also reported that RIP3 regulates early reperfusion injury via oxidative-stress- and mitochondrial-activity-related effects, rather than via cell loss due to necroptosis [54]. Furthermore, it was confirmed [40] that oxidative-stress-related subcellular deteriorations are associated with cardiac dysfunction.

Another example of a co-existing cell death type under conditions of HF might be related to a link between NADPH oxidase 2 (NOX2) activation, one of the non-canonical signaling pathways of necroptosis, and excessive production of ROS followed by plasma lipid peroxidation, a reliable marker of ferroptosis [42]. An interlink between ferroptosis and pyroptosis has also been demonstrated in a pressure overload model of HF [45]. These novel cell death modes resembling phenotypes of necrosis directly/indirectly associated with inflammation are linked with cardiac damage and repair, resulting in collagen-based scar formation and interstitial fibrosis leading to impaired cardiac function and morphology. Thus, interventions targeting the molecules of pro-inflammatory response might be effective cardioprotective tools in the management of HF [44].

## 4. Management of Heart Failure

One of the main strategies for combating HF is to treat the primary disease—its most common reason (IHD and AMI). Early restoration of blood flow in the ischemic myocardium leads to the burst of ROS production and intracellular Ca^2+^ overload as a major cause of myocardial I/R injury, rapid normalization of pH upon reperfusion, and oxidative stress resulting in the opening of the mitochondrial permeability transition pores (mPTPs) and triggering of cell death [55]. In addition, deterioration of functional and structural properties of the myocardial capillary network followed by dysfunction of endothelial cells contributes to the severity of I/R injury and further development of HF by reducing the availability of NO and limiting blood flow [56]. Due to the complexity of molecular mechanisms involved in the development of HF, its management should be multi-targeting rather than focusing on only one pathophysiological mechanism. 

However, despite advances in pharmacotherapy, interventional cardiology (primary percutaneous coronary intervention (PPCI)), and coronary artery bypass grafting (CABG) surgery, there is a substantial requirement of searching for novel approaches that could specifically address the repair of damaged and/or lost myocardium because of a very low ability of regeneration in terminally differentiated adult cardiac cells [57]. 

## 5. Innate Cardioprotection

### 5.1. Short-Term Cardiac Endogenous Protection—Ischemic “Preconditioning”

It has been revealed in numerous experimental and clinical studies that brief periods of moderate stress trigger short-term or longer-lasting adaptive processes in the heart, which ensue in greater resistance against prolonged I/R injury. The phenomenon of ischemic preconditioning (IPC) described in [8] is based on the principle that short episodes of cardiac exposure to ischemia increase heart tolerance to subsequent sustained ischemic stress. This very robust form of intrinsic cardioprotection that has been observed in all animal species, including humans [9], manifests in a delay of necrotic and apoptotic processes in cardiomyocytes [58], reduction in lethal arrhythmias [59], and improved post-I/R functional recovery [60]. In 1992, Shizukuda et al. [61] extended these findings by demonstrating that a 5-minute coronary artery perfusion with hypoxic blood (~5% O_2_) was as cardioprotective as a 5-minute coronary artery occlusion. This finding suggested that cardioprotection by IPC may be elicited primarily by insufficiency in oxygen delivery during coronary artery occlusions. Clinical studies have also revealed that patients with unstable angina pectoris (UAP) had better prognosis in subsequent AMI than patients without UAP [62]. Therefore, UAP is considered to be a clinical analogue of IPC. Moreover, IPC applied prior to PPCI improves patient outcomes, in addition to the prognosis among patients with heart transplantation [63]. Unfortunately, the feasibility of clinical implementation of IPC is reduced due to technical requirements (chest opening to obtain access to coronary arteries), unpredictable occurrence of AMI (although it might be expected), and a short-term (2–3 h) duration of cardioprotection. Thus, the application of IPC in patients is limited to elective interventions including PPCI or CABG surgery [7,63]. However, unlike the early phase of IPC, its delayed phase, the so-called “second window of protection”, that appears 24 h after the first preconditioning (PC) stimulus, is associated with the activation of multiple stress-related genes; synthesis of protective proteins is more relevant from a clinical point of view, since it lasts 3–4 days [64,65].

#### Ischemic “Postconditioning”

In addition to IPC, a strategy that can modify reperfusion-induced deleterious events through brief reperfusion (reoxygenation)/reocclusion (hypoxia) episodes after long-term ischemia, termed as ischemic postconditioning (IpostC), may have better potential for use in AMI management among humans [66]. A beneficial effect of IpostC has also been revealed in cultured cardiomyocytes exposed to long-term hypoxia, when brief episodes of reoxygenation–hypoxic postconditioning (HpostC) led to improved cell survival following persistent reoxygenation [67]. Protective anti-infarct and antiarrhythmic effects of HpostC have also been demonstrated in ex vivo Langendorff-perfused rat hearts [68]. Both pre- and postconditioning have equally positive impacts on myocardial susceptibility to I/R injury. Both reduce the infarct size and severity of reperfusion arrhythmias, attenuate endothelial dysfunction, inflammatory, and apoptotic processes, as well as improving the recovery of left ventricular function. However, some aspects of their mechanisms are different, relating to the timing of their application. While IPC can stimulate an adaptive response that increases tissue resistance against long-term ischemia via moderate generation of ROS during an IPC protocol [69], the mechanisms of IpostC induced by brief I/R or hypoxia–reoxygenation cycles do not involve this effect [66] and reduce oxidative stress and lipid peroxidation during recovery of oxygen supply [70]. In contrast to IPC, IpostC may have wider clinical applications because it can be applied during reperfusion after angioplasty, stenting, cardiac surgery, and transplantation.

### 5.2. Other “Conditioning” Interventions

Importantly, a powerful protection against I/R injury can be rendered by several other “conditioning” approaches that do not require invasive interventions or special techniques. Such interventions include adaptation to hypoxia, modulation of temperature, PC of a distant organ (remote PC), exercise-induced PC, or pharmacological PC through applying substances that mimic the processes of PC (Figure 1). 

For instance, early and rapid mild hypothermia (32–35 °C) has been reported as potent intervention in reducing myocardial infarct size, post-I/R contractile dysfunction, and attenuating left ventricular remodeling in various animal species and models [71]. It was shown that mild hypothermia reduced ROS production and preserved mitochondrial respiration in adult rat ventricular cardiomyocytes, subjected to simulated ischemia at 32 °C (compared to 38 °C) and attenuated oxidative stress in an in vivo myocardial ischemia model in rabbits [72].

Another example of an alternative “conditioning” is a short-term caloric restriction that was reported to improve postinfarct left ventricle (LV) function accompanied by a reduction in serum BNP, a decrease in LV proapoptotic activation, and an increase in mitochondrial biogenesis among rats and mice LV [73].

#### 5.2.1. Remote Ischemic “Preconditioning”

Powerful protection against I/R injury can be induced by “remote” preconditioning (RPC), in which ischemia (or another adaptive stimulus) of any organ confers protection to other, distant organs/tissues [41,74]; this appears to have potential from a clinical point of view (Figure 1). The protocol of RPC performed on limbs was termed as “limb ischemic preconditioning” (LIPC) [75] and started to be implemented in both clinical situations and in animal experiments. In particular, this noninvasive and easily applicable mode of RPC effectively reduced the size of infarction and incidence of reperfusion-induced ventricular arrhythmias, as well as improving the postischemic recovery of heart contractile function [76,77]. It has started to be used in clinical conditions, e.g., among patients with AMI as an adjunct therapy during PPCI or CABG surgery [78,79,80,81], as well as in surgery of congenital heart defects and in surgery for young children [82,83]. Importantly, a protocol of RPC was applied among humans using a pressure cuff (placed on the upper extremity), and three cycles of 5 min inflation (200 mmHg)/5 min deflation, which successfully attenuated I/R-induced endothelial dysfunction in forearm blood vessels and improved post-I/R forearm blood flow [56,84] (Figure 1). Moreover, this intervention could be applied in the settings of pre-, per-, or postconditioning, and not only in one pre-ischemic setting of several bouts of limb ischemia (Figure 1). In addition, it could be applied as repeated cycles of limb I/R in the long term that increase the efficiency of RPC [85]. Repeated RPC reduced the extent of ventricular remodeling and mortality over 28 days after AMI in a rat in vivo model [56] and increased endothelium-dependent vasodilatation among healthy humans and patients with chronic HF. The immediate protection conferred by RPC also involved mitigation of I/R-induced depression of mitochondrial membrane fluidity and a trend for better preservation of mitochondrial state 3 respiration [86]. However, despite several promising clinical studies, large clinical trials have failed to demonstrate the benefits of RPC, and the reasons for that were carefully analyzed [87]. 

It has been generally accepted that the reasons for unsuccessful results are, to a major extent, related to comorbidities/comedications and/or to the confounding factors present among patients. Main comorbidities include chronic stress, hypertension, hyperglycemia, diabetes, and hyperlipidemia [32,33,34,35], while age and gender are known as confounding factors [88]. In addition to the risk factors of CVD, gender- and age-related differences play an essential role in the heart response to conditioning interventions, such as RPC. Our recent studies also demonstrated age-dependent blunting of the RPC-induced cardioprotection in the hypertensive (SHR) male rats [38]; the normotensive rats started during the period of their maturation [30].

In addition to RPC, other noninvasive adaptive interventions that increase cardiac ischemic tolerance include adaptation to chronic hypoxia (stay in the atmosphere with reduced O_2_) [89] and its acute form—hypoxic PC [90] or HpostC [68]. Further intervention increasing the heart’s own resistance to ischemia similar to RPC is exercise-induced PC [91,92,93] (Figure 2).

#### 5.2.2. Exercise-Induced “Conditioning”

Exercise-induced conditioning is currently considered to be a natural, noninvasive form of cardioprotection, sharing its mechanisms with RPC [94].

These noninvasive protective interventions (including RPC, adaptation to hypoxia, and exercise) have been investigated in detail in the final end-effector phase of a protective cascade of events; this occurs during I/R injury, when numerous common cellular mechanisms [10], especially those related to mitochondrial function [55,95,96], have been reported.

On the other hand, relatively less attention has been paid to the triggering mechanisms and processes activated immediately after the application of an adaptive stimulus and transfer of the signal to its target. It has been shown that exercise training also acts as a stimulus attenuating Ca^2+^ cycling disorder and acts through pathways similar to IPC and RPC [92,93]. 

It is, therefore, assumed that triggering several forms of cardioprotection (RPC, exercise PC) involves similar neuronal signaling (parasympathetic and sympathetic activation) [97]. The signal can be further transmitted through the complex neuronal and humoral pathways [80] and through systemic response (suppression of inflammation, oxidative stress, and changes in the gene expression) [84]. In accord, in mice subjected to RPC 24 h prior to myocardial I/R, anti-inflammatory cytokine interleukin-10 (IL-10) has been found to be upregulated [7] (Figure 2). It is suggested that the humoral pathway is associated with the activation of cell survival cascades RISK (reperfusion injury salvage kinase)–PI3K/Akt-GSK3ß-mPTP and SAFE (survival activating factor enhancement)–TNF-α-IL-10-STAT3-mitoKATP (mitochondrial ATP-dependent K^+^ channels) [41,98] (Figure 3). It is expected that these pathways can also be activated by exercise stimuli.

Many studies have investigated the cardioprotective effect of exercise in pathological conditions. Based on the data from the literature, the cardioprotective effect of exercise training is maintained in states such as diabetes, HF, and myocardial hypertrophy. Yang et al. [99] demonstrated that four weeks of exercise training provide acute and sustained cardioprotection against isoproterenol-induced cardiac hypertrophy, and the increase in eNOS signaling molecules contributed to heart protection. Similarly, Calvert et al. [100] have shown the cardioprotective effects of exercise to be mediated by alterations in the phosphorylation status of eNOS, leading to an increase in cardiac NO and NO metabolite (nitrite and nitrosothiols) levels. These alterations are induced in part by increased wall shear stress during exercise and in part by increased β3-adrenergic receptor (AR) stimulation during exercise by circulating levels of catecholamines (Figure 4). Moreover, NO metabolites can be stored in the heart, providing the source of bioavailable NO during myocardial ischemia [100]. Additionally, Wang et al. (2017) showed that the cardioprotective effect of exercise in mice with HF (improved cardiac systolic function and alleviated LV chamber dilation, cardiac fibrosis, and hypertrophy) is associated with the activation of the β3-AR-nNOS-NO pathway [101]. On the other hand, Kleindienst et al. [102] reported deficient β3-AR-eNOS-NO signaling in exercised obese diabetic mice that cannot protect the heart against I/R. However, exercise is still an effective cardioprotective strategy in this model. Other studies revealed that exercise training protects against pathological cardiac hypertrophy through the activation of the PI3K/Akt pathway [103,104], which can be mediated through β3-AR stimulation [105] (Figure 4). Moreover, the high-intensity interval training increases the expression of VEGF, TFAM, PGC-1*α*, and mir-126 genes in the heart tissue of male Wistar rats, which probably reduces cardiac tissue injury by increasing mitochondrial biogenesis and angiogenesis following isoproterenol treatment [106]. In streptozotocin-induced diabetes, exercise (swimming and training in rats) is effective in attenuating myocardial fibrosis and contractile dysfunction [107], reducing the level of TNF-α, and increasing capillary density [108]. Also, in humans, an exercise program improved diastolic function in men diagnosed with diabetes mellitus [109]. Finally, the study of Börzsei et al. [110] demonstrated the prominent role of voluntary exercise in mitigating aging-related cardiovascular dysfunction in both female and male rats. 

However, future studies are required to determine the proper intensity, duration, and frequency of exercise to prevent cardiac injury.

## 6. Intracellular Mechanisms Involved in Cardioprotection of “Conditioning” 

### 6.1. Micro-RNAs

Several pathophysiological mechanisms of heart injury, as well as the mechanisms of protection against I/R, are regulated by less-known molecules, such as small noncoding RNA (like microRNA or miRNA), which are small segments of RNA (22–24 nucleotides) that have important regulatory roles in cell biology and cardiovascular pathophysiology [111]. MiRNAs act as a posttranscriptional regulator of the expression of protein-coding genes through sequence-specific recognition of the 3′ or the 5′ untranslated regions (UTRs) of mRNAs. The binding to the 3′ UTR lowers mRNA levels by decreasing the stability of the mRNA, leading to increasing its degradation or repressing translation [112]. MiRNAs also bind to the promoter region in the 5′ UTR of mRNAs, leading to the repression or activation of the translation [113]. MiRNAs are released from cells to blood by binding to proteins or encapsulated in vesicles and, in that state, are transported to target cells. Thereby, miRNAs seem to be well-suited substances to mediate the effects of RPC [114]. 

It has been demonstrated that the effects of miRNAs differ between different forms of heart injury [79,115]. Their role in the mechanisms of oxidative-stress-related diseases, including myocardial ischemia, has been reviewed by Kura et al. [111]. On the other hand, miRNAs have also been shown to be involved in the mechanisms of various protective “conditioning” interventions. Thus, while ischemic PC is associated with a rise in miRNA-1 and miRNA-21 in the rat heart, RPC and IpostC reduce miRNA-1 in the myocardium and have no effect on miRNA-21 [20]. Out of the significant number of those miRNAs, miRNA-144 in particular participates in the cardioprotective effect of RPC, and it is also associated with enhanced RISK cascade–PI3K/Akt-GSK3ß-mPTP signaling [116]. It has also been demonstrated that in patients undergoing CABG with prior RPC; miRNA-338-3p levels in right atrial tissue samples were higher than those of the controls [74]. 

In another study, the authors of [59] investigated the antiarrhythmic effects of IPC and postconditioning by measuring circulating miRNAs that are related to cardiac conduction in pigs. They observed significantly lower levels of circulating miRNA-1, miRNA-208, and miRNA-328 after acute myocardial infarction with the postconditioning as compared to animals in the group with acute myocardial infarction only. Minghua et al. [117] observed the protective effect of RPC mediated by miRNA-24 after I/R injury. In this study, authors report that the addition of miRNA-24 to H9c2 cells with induced ischemia and RPC revealed a decrease in oxidative stress and apoptosis by downregulating the expression of proapoptotic protein Bim in H_2_O_2_-treated H9c2 cells. Other cardioprotective miRNAs connected with PC or postconditioning conditions with previous I/R injury are miRNA-487b, miRNA let-7b, miRNA-208, and miRNA-125b [118].

#### Long Noncoding RNAs

One of the most recently identified classes of noncoding RNAs are long noncoding RNAs (lncRNAs) (lncRNAs > 200 nucleotides up to 100 kilobases), which—compared with miRNAs—have complex roles, as they can regulate gene expression through post-transcriptional, translational, and epigenetic modes of action [119]. Many lncRNAs can stimulate or inhibit cell death, including autophagy in ischemic hearts by sponging specific miRNAs and/or regulating related signaling pathways. The cardiac-specific lncRNA Zinc finger antisense 1 (ZFAS1) is overexpressed in AMI and promotes cell death and myocardial injury via downregulation of miR-150 and activation of C-reactive protein (CRP) [120]. Regulator of reprogramming (ROR) and KQT-like subfamily, member 1 opposite strand/antisense transcript 1 (KCNQ1OT1) lncRNAs are highly expressed in patients with I/R injury and in hypoxia-reperfusion-treated cardiomyocytes and lead to apoptosis through regulation of p38 mitogen-activated protein kinase (MAPK) and nuclear factor kappa B (NF-κB) signaling pathways [121]. 

However, only a few studies have illustrated the role of lncRNAs in myocardial ischemic conditioning. It has been shown that morphine postconditioning attenuates myocardial I/R injury in a rat model through upregulation of the lncRNA urothelial carcinoma-associated 1 (UCA1), which downregulated miR-128 and expression of autophagy markers [122]. Furthermore, other studies illustrate the regulatory role of lncRNA H19 in myocardial I/R injury and cell death; this was performed through targeting miR-103/107 or miR-877-3p [121,123,124]. Authors have demonstrated that lncRNA H19 is upregulated in H_2_O_2_ PC-treated H9c2 cells in hypoxia PC-treated neonatal rat cardiomyocytes and in vivo in murine hearts subjected to IPC,; additionally, this protects ischemic heart through posttranscriptional regulation of nucleolin protein. However, no data about the role of lncRNAs in cardioprotection through RPC have been published so far. Thus, the role of miRNAs in the mechanisms of RPC or exercise-induced PC has not been sufficiently explored so far.

### 6.2. Peroxisome Proliferator-Activated Receptors

Nuclear peroxisome proliferator-activated receptors (PPARs) are a family of nuclear receptors consisting of three known isoforms (α, β/δ, γ). Their main role in the organism is the regulation of genes involved in the processes of metabolism and energy production in the heart [125]; they additionally have a role during different pathological conditions in the cardiovascular system, including I/R, HF, and metabolic disorders, such as diabetes. Moreover, it has been shown that the ligands of these transcription factors may induce pleiotropic PC-like lipid-independent genomic and non-genomic effects, including anti-apoptotic and anti-inflammatory effects (so-called pharmacologically induced PC with subsequent myocardial protection against I/R injury) [126,127]. It has also been demonstrated that PPARα and PPARγ isoforms play a crucial role in the protective mechanisms of an in vivo rabbit model of RPC associated with activation of 15d-prostaglandin J2 and increased iNOS expression [77]; meanwhile, both anti-infarct protection and molecular changes were abrogated by PPARα and PPARγ inhibitors. Furthermore, myocardial protection afforded by a mouse RPC model has been found to be mediated via the PI3K/Akt/GSK3β signaling pathway, activation of which was associated with nuclear accumulation of β-catenin and the upregulation of its downstream targets: E-cadherin and PPARβ/δ these are involved in cell survival (IS-limitation, improved contractile function, and reduced apoptosis) [116]. Moreover, cardioprotective effects of PPARβ/δ activation against I/R in rat hearts are associated with ALDH2 upregulation, amelioration of oxidative stress, and preservation of mitochondrial energy production [128]. The role of PPARs in the mechanisms of RPC and long-lasting adaptive processes induced through exercise training and hypoxia, as well as in chronic processes leading to the development of HF, remains incompletely explored.

Currently, it is believed that transmission of the protective signal to the target organ is multifactorial, requiring a combination of humoral, neuronal, and systemic mechanisms.

#### Role of Mitochondria in Cardioprotective Mechanisms

Over the recent years, the view on mitochondria in the heart as a cellular powerhouse providing adenosine triphosphate (ATP) supply needed to sustain contractile function, basal metabolic processes, and ionic homeostasis has changed radically. At present, it is known that dysfunctions of these organelles are essential in the development of many diseases, including cardiovascular diseases [86,129,130,131]. Moreover, mitochondria are a very promising target of endogenous strategies that are essential in protecting the myocardium from acute I/R injury. These strategies, including IPC, RPC, and other noninvasive adaptive approaches, provide a similar effect of protection [132]. Preservation of mitochondrial membrane function and respiratory properties has been shown to be an important mechanism of cardioprotection [132,133]. Mitochondria are involved in attenuation of processes, leading to cell death because end-effector systems (mitoKATP, mPTP) become localized in mitochondria [55,95,96]. Regulation of mPTP seems to be an important part of the mechanisms that maintain the energy equilibrium of the heart under pathological conditions. There are two ways in which the mPTP opens (Figure 5). Pathological processes, as well as I/R injury, induce prolonged mPTP opening [134,135]. Prolonged mPTP opening increases the permeability of the inner mitochondrial membrane (IMM) and allows the entry of metabolites into the mitochondrial matrix, which in turn leads to the decrease in membrane potential of the mitochondrial membrane, the disconnection of the respiratory chain, the cessation of ATP synthesis in the mitochondria, and eventually mitochondrial swelling, leading to the rupture of the outer mitochondrial membrane (OMM), which releases cytochrome c (Cyt c) and causes cell death [136,137,138]. PC induces a cardioprotective effect through inhibiting the opening of mPTP [139,140]. Transient opening of the mPTP belongs to the physiological processes that mitochondria use in the healthy functioning of the cell [141] associated with a transient increase in ROS as signaling molecules and increased membrane fluidity [132,142]. It is thought that the transient opening of the mPTP may regulate Ca^2+^ in the cytosol when Ca^2+^ overload occurs. Opening of mitoKATP induces a moderate increase in ROS production, which activates protective signaling, preventing reperfusion-induced mPTP opening and loss of mitochondrial membrane integrity. That is followed by disruption of the integrity of the mitochondrial membrane and the release of proapoptotic molecules (e.g., Cyt c), activating mechanisms of cell death [132,143]. Thus, simulating the effect of RPC through pharmacological modulators of mPTP opening using drug cyclosporin A [144] appears to be a promising approach; this has already been applied in patients with AMI prior to angioplastic intervention [145,146]. It can be mentioned that such modulation of mPTP opening has also been suggested to underlie, in part, the cardioprotection of necrostatin-1, a drug that inhibits necroptosis, because its infarct size-reducing effects failed in cyclophilin-D-deficient mice (cyclophilin-D is a key component of mPTP) [147]. Although many questions are unresolved in this regard, including the precise role of mitochondria in necroptosis execution, these findings, as well as the data about the anti-necroptotic action of IPC [148], support the theory of cardioprotection by limiting this cell death due to mitochondria modulation.

## 7. Concluding Remarks

In the coming decades, the incidence of HF is predicted to rise; this is the case due to the aging of the population. An aging population is associated with the development of comorbidities, such as hypertension, diabetes, obesity, and others. However, increasingly longer survival after AMI has been noted, due to successful cardiac surgery and interventional cardiology, as well as pharmacotherapy. Despite certain progress in the treatment of AMI, long-term prognoses are still not encouraging. Thus, detailed elucidation of molecular mechanisms regulating the function of healthy and diseased myocardium and adaptive processes in cardiac cells is crucial in the battle against pathological processes, including myocardial ischemia. This is the case not only in experimental conditions; it is essential for the survival of human beings because it will ensure an increase in their resistance to I/R injury. In particular, noninvasive adaptive interventions appear to be promising as potential tools for rendering remarkable, cost-effective anti-ischemic protection among people suffering from clinical conditions. Their implementation as an adjunct therapy among patients with chronic IHD or AMI may help to optimize their treatment, delay the onset of critical events in the development of HF, and delineate further prognoses.

## Figures and Tables

**Figure 1 ijms-24-16497-f001:**
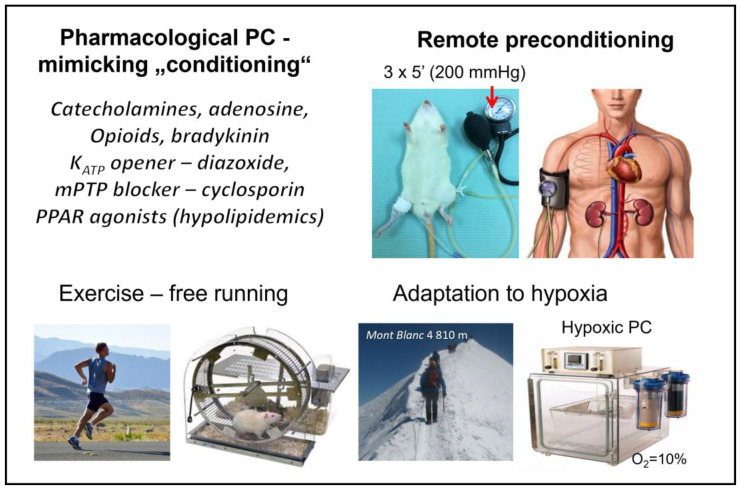
Noninvasive cardioprotective interventions trigger mechanisms of intrinsic cardioprotection. Different forms of “conditioning” approaches that do not require invasive interventions or special techniques and confer protection against I/R injury. Detailed information and abbreviations are provided in the text.

**Figure 2 ijms-24-16497-f002:**
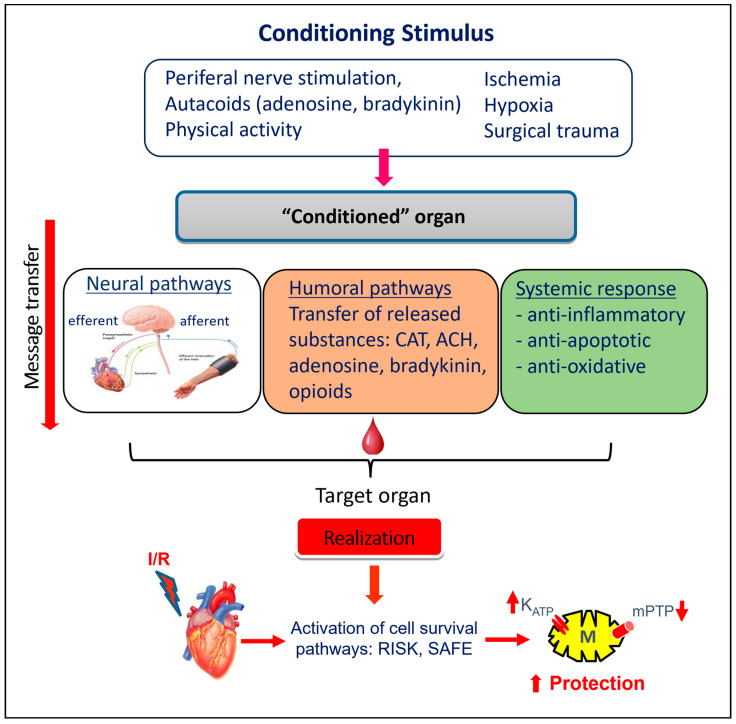
Simplified scheme of triggering of cardioprotection and transmitting the signal to the distant organ and target end-effectors in the heart cells. Three theories of message transfer from the “conditioned” organ to a one—neural, humoral (neurohumoral) pathways, and systemic response. Detailed information and abbreviations are provided in the text.

**Figure 3 ijms-24-16497-f003:**
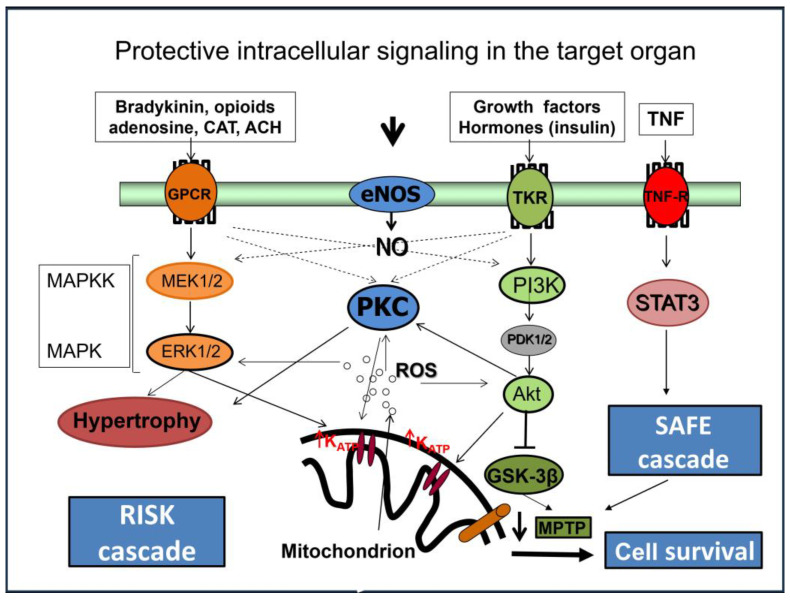
Intracellular mechanisms involved in cardioprotection of conditioning. Protective signaling in the target organ, including activation of intracellular RISK and SAFE cascades, leads to cell survival and protection. Detailed information and abbreviations are provided in the text.

**Figure 4 ijms-24-16497-f004:**
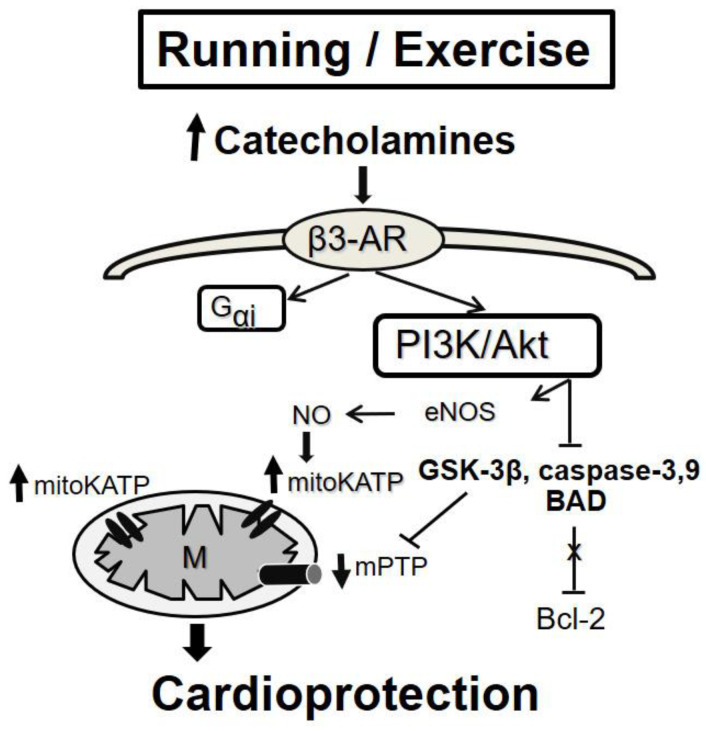
Potential scheme of exercise PC-induced cardioprotection. The cardioprotective effect of exercise can be associated with activating the β3-adrenergic receptor signalization. Detailed information and abbreviations are provided in the text.

**Figure 5 ijms-24-16497-f005:**
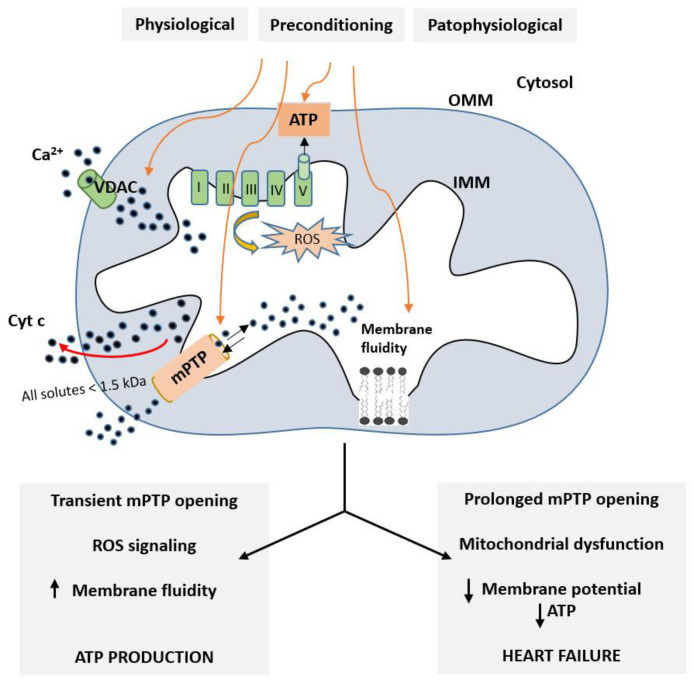
mPTP signaling pathways in preconditioned and pathological myocardium. There are two ways in which the mPTP opens—transient and prolonged. VDAC—voltage-dependent anion channel. For more details, see [132].

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
