# Peer review of "Is Intrinsic Cardioprotection a Laboratory Phenomenon or a Clinically Relevant Tool to Salvage the Failing Heart?"

_ijms, 2023, doi:10.3390/ijms242216497_

Round 1
Reviewer 1 Report
Comments and Suggestions for Authors
The review article by Ravinestá and colleagues is well described, I enjoyed reading it very much. However, the information contained in some paragraphs is very dense so it could be lightened a bit (e.g., L235-260, L303-335, L427-469). Overall, the article contains relevant and current information in the field of heart failure with important points to study in future years. I leave the authors some comments that could be considered.
· There are stylistic and grammatical issues that could be corrected (e.g., L30, L68, Figure 1, among others).
· Define for the first time the abbreviations used (e.g., RIP3, MKL, TNF, csp-8, among others).
· L154. A connector such as "and" is missing between ...ROS production, intracellular Ca2+ overload....
· In general, figures could be supplemented with a brief figure caption.
· The reference for L362 seems to be missing in the journal format. Please revise.
· Revise the references in terms of year of publication (I would only consider those that are no more than 5 years old) as well as avoid citing review articles and focus only on original and experimental manuscripts.
Comments on the Quality of English LanguageThe English language is adequate, only minor editing is required.
Author Response
Please see the details in the attachment.

Reviewer 2 Report
Comments and Suggestions for Authors
The review by Ravingerova and colleagues emphasizes that despite advancements in cardiovascular treatments, the prevalence of ischemic heart disease leading to heart failure remains high, especially among the elderly and females. The review focuses on the concept of ischemic preconditioning (IPC), which shows promise in protecting the heart from damage due to moderate stress. The review also summarized non-invasive approaches, like adaptation to hypoxia and exercise-induced preconditioning, to enhance cardiac tolerance. Although the molecular mechanisms are not fully understood, these approaches hold potential as innovative strategies to combat heart failure. This is a well-written comprehensive review with the highlight of important public health issues. Some minor revisions are suggested.
1. The review would benefit from a more extensive discussion regarding the potential clinical translational value of the experimental findings and an explanation of the technical challenges that impede their implementation in clinical settings.
2. Given the crucial role of IR injury in heart transplantation, it would be valuable to delve into discussions about strategies aimed at preventing IRI in the context of heart transplants and promoting cardioprotection.
3. The incorporation of animal models of investigating heart IR injury and cardioprotection is commendable and adds depth to the review.
Comments on the Quality of English Languageminor editing
Author Response

(The authors gave the same response as above.)
